# Development of the Volatile Fingerprint of Qu Aurantii Fructus by HS-GC-IMS

**DOI:** 10.3390/molecules27144537

**Published:** 2022-07-15

**Authors:** Cuifen Fang, Jia He, Qi Xiao, Bilian Chen, Wenting Zhang

**Affiliations:** 1Zhejiang Institute for Food and Drug Control, Hangzhou 310052, China; fcf0507@126.com; 2NMPA Key Laboratory for Quality Evaluation of Traditional Chinese Medicine (Traditional Chinese Patent Medicine), Hangzhou 310052, China; 3Hangzhou Zhongce Vocational School Qiantang, Hangzhou 311228, China; hejia@zcmu.edu.cn; 4College of Pharmaceutical Sciences, Zhejiang Chinese Medical University, Hangzhou 310053, China; 201911113711455@zcmu.edu.cn

**Keywords:** Qu Aurantii Fructus, HS-GC-IMS, fingerprint, Aurantii Fructus, adulterants

## Abstract

Volatile components are important active ingredients of Rutaceae. In this study, HS-GC-IMS (headspace-gas chromatography-ion mobility spectrometry) was used to study the volatile compounds of Qu Aurantii Fructus, and a total of 174 peaks were detected, 102 volatile organic compounds (131 peaks) were identified. To compare the volatile compounds of Qu Aurantii Fructus with its similar medical herb, Aurantii Fructus, and their common adulterants, principal component analysis (PCA) and cluster analysis (CA) were performed based on the signal intensity of all the detected peaks. The results showed that Qu Aurantii Fructus and Aurantii Fructus (*Citrus aurantium* L.) were clustered into one group, while their common adulterants could be well distinguished in a relatively independent space. In order to distinguish Qu Aurantii Fructus from Aurantii Fructus, the peaks other than the average intensity ±2 standard deviation (95% confidence interval) were taken as the characteristic components by using the Gallery Plot plug-in software. Additionally, the fingerprint method was established based on the characteristic compounds, which can be used to distinguish among Qu Aurantii Fructus, Aurantii Fructus and their common adulterants quickly and effectively. We found that the characteristic components with higher content of Qu Aurantii Fructus were nerol, decanal, coumarin and linalool. This study provides a novel method for rapid and effective identification of Qu Aurantii Fructus and a new dimension to recognize the relationship between Qu Aurantii Fructus and Aurantii Fructus.

## 1. Introduction

Qu Aurantii Fructus is recorded in the 2015 edition of the processing standard of traditional Chinese medicine in Zhejiang Province [1]. It is the dried, immature fruit of *Citrus changshan*-*huyou* Y.B. Chang, which is harvested in July when the fruit is still green. It has the function of regulating qi width and relieving flatulence. It is used to relieve chest and hypochondriac qi stagnation, fullness and pain, retention of food accumulation, phlegm and internal stagnation; it is often used to treat diseases such as organ ptosis. Qu Aurantii Fructus is mainly produced in Quzhou City, Zhejiang Province, which is one of the “New Zhe-ba-wei”. Studies on chemical constituents show that Qu Aurantii Fructus mainly contains flavonoids [2,3,4], triterpenes [5], phenolic acids [6], steroids [6], and coumarins [2]. Modern pharmacological studies show that Qu Aurantii Fructus have pharmacological activities such as lung injury protection [7,8], liver protection [9,10], antioxidation [11], blood sugar lowering [12], anti-microbial [13], and so on.

In addition to Qu Aurantii Fructus, there are more medicinal plants of the *Citrus* in the family Rutaceae, for example, Aurantii Fructus, which is recorded in the Chinese Pharmacopoeia 2020 edition [14], the source of which is *Citrus aurantium* L. and its cultivated variants. The common cultivated variants in the market are *Citrus Aurantium* ‘Huangpi’, *Citrus aurantium* ‘Daidai’, *Citrus aurantium* ‘Chuluan’, and *Citrus aurantium* ‘ Tangcheng’, *Citrus aurantium* cv. Xiucheng [15]. Both being the immature fruit of the citrus, Qu Aurantii Fructus, Aurantii Fructus are very similar in appearance after processing and more difficult to distinguish. Additionally, they have been taken for the same in some markets. In addition to this, there are some close relatives of Rutaceae, such as *Citrus*
*wilsonii* Tana-ka, *Citrus reticulata* ‘Unshiu’, *Citrus sinensis* (Linn.) Osbeck, and are often mixed as Qu Aurantii Fructus and Aurantii Fructus, making the use of Qu Aurantii Fructus in the market more confusing [16].

At present, the quality control of Qu Aurantii Fructus and Aurantii Fructus mainly focuses on flavonoids [17,18,19,20,21,22,23]. Some scholars have studied the fingerprint of flavonoids, and found the problem that Qu Aurantii Fructus cannot be distinguished from some sources of Aurantii Fructus [24]; meanwhile, flavonoids of different species of Aurantii Fructus are very different. For example, the content of flavonoids in Aurantii Fructus (*Citrus aurantium* ‘Chuluan’) is very low, which cannot even meet the requirements of the standard [21]. Thus, this kind of differentiating method is ineffective.

The volatile compounds in the fruit of Rutaceae are high in content and have strong specificity. It is reported that the volatile is an important active compound of Qu Aurantii Fructus and Aurantii Fructus [13,25]. The content of volatile is used as the quality control indicator in European Pharmacopoeia 10.0 and Japanese Pharmacopoeia XVII [26,27], which accounts for the importance of volatile in quality control. There are literatures which used GC-MS (gas chromatography–mass spectrometry) to study the volatile compounds of Aurantii Fructus, and the main component was found to be limonene, with a relative percentage content of more than 50% [25,28,29]. However, there is no study on the volatile compounds of Qu Aurantii Fructus and a systematic comparison between them.

HS-GC-IMS is a new technique developed in recent years for the detection of aromatic compounds, by which substances can be separated in two dimensions by GC and IMS drift tubes [30,31,32,33,34]. The method does not require complex sample pretreatment, and the sample can be directly injected after crushing, which has the advantages of environmental friendliness, high sensitivity and short analysis time [23,35]. HS-GC-IMS has a good application in the detection of flavor components in the food field, and has been increasingly widely used in the field of pharmaceutical research in recent years. Jia He used HS-GC-IMS method for the identification of adulterated inferior products in Ophiopogon, and this method showed a higher degree of identification [36].

In this study, the volatile compounds of Qu Aurantii Fructus were studied by HS-GC-IMS. By comparing the volatile compounds of Qu Aurantii Fructus with its similar medical herb, Aurantii Fructus, and their common adulterants, the relationship between Qu Aurantii Fructus and Aurantii Fructus was found based on statistical analysis, and the fingerprint of characteristic components fitted by the Gallery Plot plug-in software was established to provide a novel reference for the quality control of Qu Aurantii Fructus.

## 2. Results

### 2.1. Volatile Compounds and Semi-Quantitative Analysis of Qu Aurantii Fructus

After the sample was analyzed by HS-GC-IMS, the data was represented by 3D topographical visualization in Figure 1, where the X axis represented the drift time relative to the reaction ion peak, Y axis represented gas phase retention time (Rt), Z axis represented ion response intensity. The n-ketones C4–C9 were used to calculate the retention index (RI) of volatile compounds as external references. Dt (RIP Rel.) was obtained by normalizing the drift time with the expected reaction ion peak (RIP). Volatile compounds were identified by comparing RI and Dt (RIP Rel.) with the GC-IMS library which contains built-in NIST (National Institute of Standards and Technology, 2014) database and IMS (ion mobility spectroscopy, G.A.S, Dortmund, Germany) database. Volatile compounds were abundant in Qu Aurantii Fructus as 174 peaks were detected. It was found that some compounds produced dimer and trimer peaks in the process of ionization, resulting in multiple peaks for those compounds. A total of 102 compounds (131 peaks) were identified by using GC × IMS Library search software. The volatile compounds in Qu Aurantii Fructus were mainly terpenoids. The detailed information is shown in Table 1. The area percentages (%) of the volatile compounds in the 8 batches of Qu Aurantii Fructus are shown in Table 1, and the box content diagrams of the main components are shown in Figure 2. The compounds with higher area percentages (%) are α-farnesene (10.4%), limonene (6.9%), γ-terpinene (6.2%), linalool (5.5%), α-terpineol (5.1%), camphene (4.5%), β-ocimene (4.4%), methyleugenol (4.2%), linalool oxide (2.9%), α-thujene (2.8%), nerol (2.3%), β-pinene (2.2%), linalyl acetate (2.1%), tricyclene (2.0%), α-terpinene (2.0%), terpinen-4-ol (1.6%), (Z)-β-farnesene (1.5%) and so on.

### 2.2. Comparative Analysis of Unique Volatile Compounds in Different Samples

All the detected peaks of Qu Aurantii Fructus and Aurantii Fructus and their common adulterants were selected for fingerprint comparison using the Gallery Plot plug-in, as shown in Figure 3 and Figure 4. We found that they have the same types of volatile components, but there are differences in the proportion. The unique components of different samples are shown as follows.

#### 2.2.1. Comparative Analysis of Volatile Compounds between Qu Aurantii Fructus and Aurantii Fructus

The comparison of the fingerprint profiles of Qu Aurantii Fructus and Aurantii Fructus was shown in Figure 3. It can be seen from the plot diagram that there are differences between Qu Aurantii Fructus and some species of Aurantii Fructus. For example, the contents of citral, benzothiazole, peak 2 and 15 of Qu Aurantii Fructus are higher, the contents of hexan-2-one, pentan-1-ol and peak 9 of *Citrus aurantium* cv. Xiucheng are higher, the contents of hexan-2-ol, butan-2-one, linalool oxide-M and geraniol of *Citrus aurantium* ‘Daidai’ are higher, the contents of α-terpineol, vanillin, peak 13 and 25 of *Citrus aurantium* ‘Chuluan’ are higher. These differential components are the basis for the identification of Qu Aurantii Fructus and Aurantii Fructus.

#### 2.2.2. Comparative Analysis of Volatile Compounds of Qu Aurantii Fructus and the Common Adulterants

Using the Gallery Plot plug-in, all the peaks of Qu Aurantii Fructus and the common adulterants were compared by fingerprint, as shown in Figure 4. The differences between the volatile compounds of Qu Aurantii Fructus and the adulterants are as follows: the relative contents of citral, benzothiazole, peak 2 and 15 are higher in Qu Aurantii Fructus; the relative contents of 3-methylbut-2-enal are higher in *Citrus wilsonii* Tana-ka; the relative contents of 2-oxopropyl acetate, 4-ethylphanol, (methyldisulfanyl) methane, 6-methyl-5-hepten-2-one, and pentan-1-ol are higher in *Citrus reticulata* ‘Unshiu’; the relative contents of acetophenone, 1-(furan-2-yl)ethanone, ethyl acetate and acetoin are higher in *Citrus sinensis* (Linn.) Osbeck. There are significant differences in volatile components between Qu Aurantii Fructus and adulterants; in particular, the three common adulterants have obvious characteristic components for identification.

### 2.3. Stoichiometric Analysis

To recognize the similarities and differences among Qu Aurantii Fructus, Aurantii Fructus, and their common adulterants, principal component analysis (PCA) and cluster analysis (CA) were performed based on the signal intensity of all the detected peaks; partial least square-discriminant analysis (PLS-DA) was performed to determine the contribution value of characteristic components.

#### 2.3.1. Principal Component Analysis (PCA)

All the detected peaks of Qu Aurantii Fructus, Aurantii Fructus and the adulterants were imported into SIMCA-P (13.0) software for principal component analysis, as shown in Figure 5. The automatic fitting of Qu Aurantii Fructus and Aurantii Fructus (*Citrus aurantium* L.) were clustered into one group (I), Aurantii Fructus (*Citrus aurantium* ‘Huangpi’, *Citrus aurantium* cv. ‘Xiucheng’, *Citrus aurantium* ‘Daidai’ and *Citrus aurantium* ‘Chuluan’) were clustered into one group (II), while three common adulterants were significantly different (III). The model test showed that R^2^X was 0.953 and Q^2^ was 0.820, which indicated that the model had good stability and predictability. The statistical results of PCA show that Qu Aurantii Fructus can be effectively distinguished from three kinds of adulterants, but is similar with Aurantii Fructus (*Citrus aurantium* L.). This is basically consistent with the same clinical efficacy of Qu Aurantii Fructus and Aurantii Fructus.

#### 2.3.2. Cluster Analysis (CA) for Qu Aurantii Fructus and Aurantii Fructus

To further validate the results of PCA analysis, all the data of Qu Aurantii Fructus and Aurantii Fructus were imported into SPSS 18.0 software for cluster analysis. According to the standard, it was found that when the distance is less than 15, Qu Aurantii Fructus and Aurantii Fructus (*Citrus aurantium* L.) were clustered into one group, and *Citrus aurantium* cv. Xiucheng and *Citrus aurantium* ‘Huangpi’ were clustered into one group, as shown in Figure 6. The statistical result of CA is consistent with that of PCA.

#### 2.3.3. Partial Least Square-Discriminant Analysis (PLS-DA)

PLS-DA was performed to determine the contribution value of characteristic components. The higher the VIP (variable importance in the projection) value of the chromatographic peak of the PLS-DA model, the greater the contribution of the chromatographic peak to the classification of the sample. The results are shown in Figure 7. Additionally, the VIP value which is greater than 1 indicates a significant effect. The results show that 23 known compounds are greater than 1.

### 2.4. Establishment of Characteristic Fingerprint of Qu Aurantii Fructus

The statistical results based on PCA and CA of all detected peaks showed that Qu Aurantii Fructus and Aurantii Fructus (*Citrus aurantium* L.) were very similar and difficult to distinguish. However, from the fingerprint profiles, there are some different components between Qu Aurantii Fructus and Aurantii Fructus (*Citrus aurantium* L.). Therefore, we tried to screen out the differential components as indicators to identify the samples.

The peaks other than the average intensity ±2 standard deviation (95% confidence interval) were taken as the characteristic components by using the Gallery Plot plug-in software, 25 characteristic compounds were screened out and fingerprints were established. As shown in Figure 8, region I is the fingerprints of different species of Aurantii Fructus, region II is the fingerprint of Qu Aurantii Fructus, and region III is the fingerprints of different adulterants. It can be seen that Qu Aurantii Fructus can be distinguished among Aurantii Fructus and different adulterants, the fingerprints of Aurantii Fructus are different, and the differences are related to the varieties. Meanwhile, it can be seen from the fingerprints that the response values of nerol, decanal, coumarin and linalool are higher in Qu Aurantii Fructus, the response values of heptanal, isopentyl hexanoate, citronellol, 2-methylbutan-1-ol and coumarin are higher in Aurantii Fructus, and the response values of acetophenone, ethyl acetate, propan-1-ol, isovaleric acid and 2-methylfuran-3-thiol in adulterants are significantly higher in adulterants, which could be used as novel components to evaluate the quality of Qu Aurantii Fructus, Aurantii Fructus and the identification of the adulterants.

## 3. Discussion

In this paper, we investigated the volatile components in Qu Aurantii Fructus, Aurantii Fructus, and their common adulterants using HS-GC-IMS. Taking Aurantii Fructus as an example, by comparing the results of HS-GC-IMS with those of GC-MS analysis reported in the literatures [28,29,37], it was found that there were differences between them. The main volatile component analyzed by GC-MS was the non-characteristic component limonene, with the relative content above 50%, and the content of other volatile components was basically below 1%. However, the volatile components analyzed by HS-GC-IMS showed that the content of limonene accounted for about 6%, linalool 7–10%, α-terpineol 4–7% and so on, the contents of more than 20 volatile components were above 1%. It is obvious that the volatile components measured by HS-GC-IMS method are more informative in terms of characteristic peaks. It is speculated that it is mainly caused by different pretreatment. When the volatile compounds are determined by GC-MS method, the sample needs steam distillation, while the sample determined by HS-GC-IMS method does not need pretreatment. The sample was grinded for direct determination, which can retain the volatile components in the sample to the maximum extent, and thus, it showed certain advantages in the identification of characteristic components.

As a similar product of Aurantii Fructus, Qu Aurantii Fructus has a very long history of use in Zhejiang Province, and its efficacy is basically the same as that of Aurantii Fructus. However, the two species currently have different legal status, as Qu Aurantii Fructus is recorded in the 2015 edition of the processing standard of traditional Chinese medicine in Zhejiang Province and can only be used in Zhejiang Province, while Aurantii Fructus is recorded in the Chinese Pharmacopoeia 2020 edition and can be used throughout China. Therefore, even if the two are similar in efficacy, they should not be mixed, and effective methods of differentiation are needed.

However, through plant taxonomic investigation, comparative study of efficacy and comparative analysis of flavonoid components [8,38,39], some scholars think that Qu Aurantii Fructus is a cultivated variety of Aurantii Fructus and can be treated without distinction.

In this paper, we compared the similarities and differences between Qu Aurantii Fructus and Aurantii Fructus in terms of volatile components, and found that they have the same types of volatile components, but there are differences in the proportion; also it was found that the volatile components in Aurantii Fructus from different sources differed significantly in the proportion. Statistical analysis (PCA and CA) was performed based on the signal intensity of all detected peaks. It was found that when the distance is less than 15, Qu Aurantii Fructus and Aurantii Fructus (*Citrus aurantium* L.) were clustered into one group, which showed that they have a good genetic relationship. In view of the similar clinical efficacy of Qu Aurantii Fructus and Aurantii Fructus, it is considered that a more comprehensive and in-depth study is required to examine whether Qu Aurantii Fructus can be used as a source of Aurantii Fructus.

The fingerprint was established based on the characteristic components screened by the software, which showed some specificity in the species differentiation. It can be intuitively seen from the fingerprints that the method can distinguish not only Qu Aurantii Fructus, but also different species of Aurantii Fructus, while more samples from accurate sources are needed for validation.

## 4. Materials and Methods

### 4.1. Materials

Eight batches of Qu Aurantii Fructus, 8 batches of Aurantii Fructus (including 2 batches of *Citrus aurantium* L., 1 batch of *Citrus aurantium* ‘Huangpi’, 3 batches of *Citrus aurantium* cv. Xiucheng, 1 batch of *Citrus aurantium* ‘Daidai’, 1 batch of *Citrus aurantium* ‘Chuluan’) and 3 batches of the common adulterants (including 1 batch of *Citrus wilsonii* Tana-ka, 1 batch of *Citrus reticulata* ‘Unshiu’, and 1 batch of *Citrus sinensis* (Linn.) Osbeck) were collected. The details of the samples are shown in Table 2. All samples were collected from their places of origin by the research group, cut in half, and dried at low temperature (40 °C).

### 4.2. HS-GC-IMS Methods

Analyses of samples were performed on a GC–IMS instrument (FlavourSpec^®^G.A.S., Dortmund, Germany), equipped with an automatic sampler unit (PAL, Analytics AG, Zwingen, Switzerland), allowing the sample to be directly injected from the headspace through a 1 mL airtight heated syringe.

Samples were ground into fine powder, and 0.5 g of fine powder were weighed and placed into a 20 mL headspace bottle. Subsequently, samples were incubated at 80 °C for 20 min at the speed of 500 rpm, then 0.5 mL of the headspace gas was automatically injected into the injector by means of a heated syringe (85 °C) in splitless mode. Then, samples were driven into a FS-SE-54-CB-1 capillary column (5% phenyl-95% dimethyl polysiloxane, 15 m in length, 0.53 mm in internal diameter, and 1μm in film thickness Restek, USA) by nitrogen (99.999% purity) at a programmed flow as follows: 2 mL/min for 2 min, increased to 100 mL/min within 20 min, and then hold for 10 min at 100 mL/min. The analytes were driven to the ionization chamber to be ionized in a positive ion mode by a tritium source(3H). The resulting ions were driven to the drift tube (98 mm in length) which operated with a constant voltage (500 v/cm) at 45 °C. Additionally, the drift gas (nitrogen, 99.999% purity) was set as 150 mL/min. Each sample was tested in triplicate.

The n-ketones C4-C9 (Sinopharm Chemical Reagent ShanghaiCo., Ltd., Shanghai, China) were used to calculate the RI of volatile compounds as external references. The drift time (RIP relative) was obtained by normalizing the drift time with the expected reaction ion peak (RIP).

Volatile compounds were identified by comparing RI and Dt (RIP Rel.) with the GC-IMS library which contains built-in NIST (National Institute of Standards and Technology, 2014) database and IMS (ion mobility spectroscopy; G.A.S; Dortmund, Germany) database. In addition, the content of each volatile compound was calculated by the normalization method based on the peak intensity.

### 4.3. Data Analysis

The data were acquired and analyzed using Laboratory Analytical Viewer (LAV) software and GC × IMS Library search software. LAV software includes two built-in plug-ins: Reporter and Gallery Plot. The Reporter plug-in was used to generate a topographic plot to visually compare the differences in 3D spectra of different samples. The Gallery Plot plug-in was used to generate fingerprint plots to visually compare the differences in peak intensities of different compounds. LAV software was used to acquire and process the IMS data and calculate the retention index (RI) of the volatile compounds using n-ketones C4-C9 as an external standard. Additionally, it was also used to filter the characteristic peaks other than the average peak intensity ±2 standard deviation (95% confidence interval) to establish the characteristic fingerprints.

Qualitative analysis was performed using GC × IMS Library search software, which contains built-in NIST (National Institute of Standards and Technology, 2014) database and IMS (ion mobility spectroscopy, G.A.S, Dortmund, Germany) database. Cluster analysis was performed by SPSS 18.0 software and principal component analysis was performed using SIMCA-P (13.0) software (MKS Data Analytics Solutions, Umea, Sweden).

## 5. Conclusions

In this study, the volatile compounds of Qu Aurantii Fructus were analyzed, and systematically compared with the components of Aurantii Fructus and their common adulterants. Based on statistical analysis, including principal component analysis (PCA) and cluster analysis (CA), the similarities and differences between Qu Aurantii Fructus and Aurantii Fructus were found. The fingerprint was established based on the characteristic components fitted by the Gallery Plot plug-in software which can be used to distinguish Qu Aurantii Fructus among Aurantii Fructus and their common adulterants effectively and quickly. The results can provide a novel reference for the quality control of Qu Aurantii Fructus and a new dimension to recognize the relationship between Qu Aurantii Fructus and Aurantii Fructus.

## Figures and Tables

**Figure 1 molecules-27-04537-f001:**
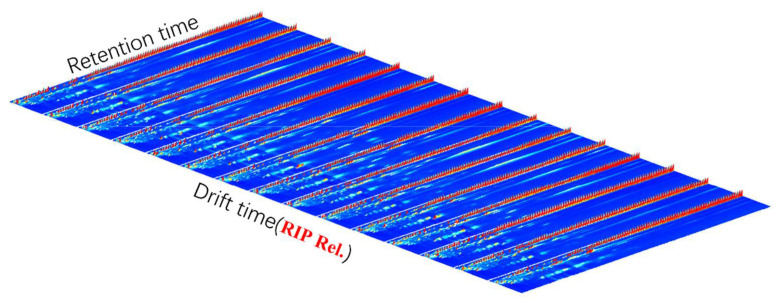
The three-dimensional spectrum of volatile compounds in Qu Aurantii Fructus.

**Figure 2 molecules-27-04537-f002:**
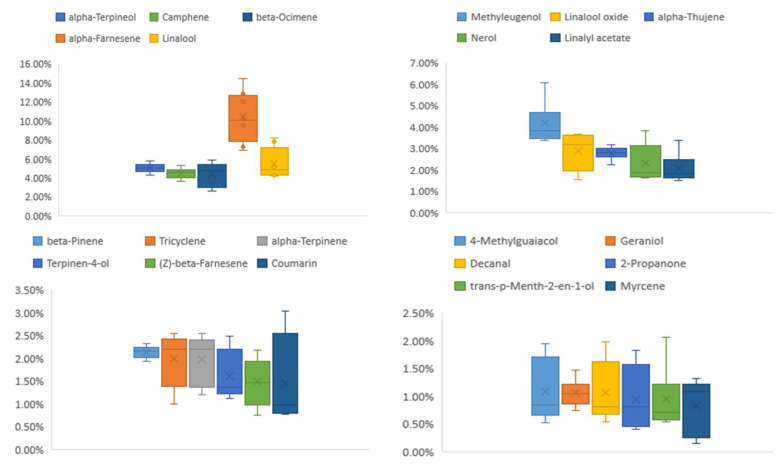
Relative contents of volatile compounds in Qu Aurantii Fructus.

**Figure 3 molecules-27-04537-f003:**
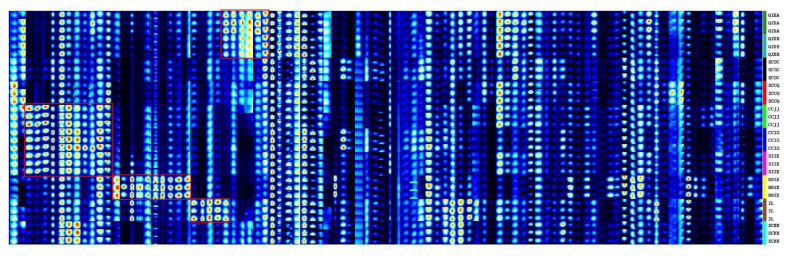
Gallery Plot diagram of volatile compounds in Qu Aurantii Fructus and Aurantii Fructus. Note: each row represents a sample (from top to bottom, 1~6 are Qu Aurantii Fructus, 7~30 are Aurantii Fructus, and 7~12 are *Citrus aurantium* L., 13~21 are *Citrus aurantium* cv. Xiucheng, 22~24 are *Citrus aurantium* ‘Daidai’, 25~27 are *Citrus aurantium* ‘Chuluan’, and 28~30 are *Citrus aurantium* ‘Huangpi’); each column represents a compound.

**Figure 4 molecules-27-04537-f004:**
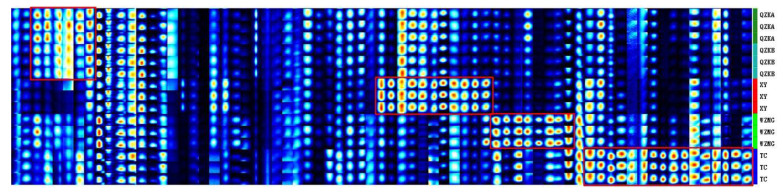
Gallery Plot diagram of volatile compounds in Qu Aurantii Fructus and its adulterants. Note: each row represents a sample (from top to bottom, 1~6 are Qu Aurantii Fructus, 7~9 are *Citrus wilsonii* Tana-ka, 10~12 are *Citrus reticulata* ‘Unshiu’, and 13~15 are *Citrus sinensis* (Linn.) Osbeck); each column represents a compound.

**Figure 5 molecules-27-04537-f005:**
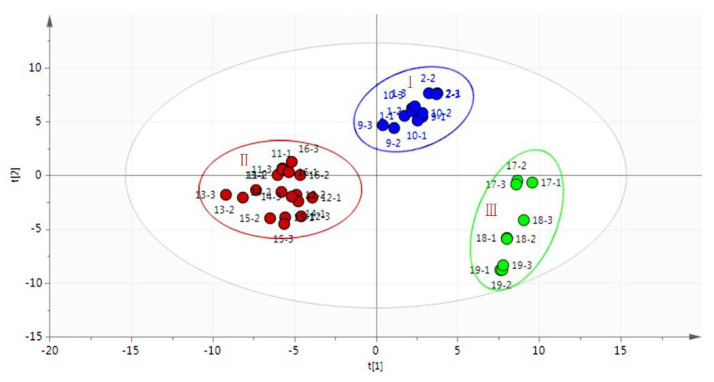
PCA analysis of volatile compounds in Qu Aurantii Fructus, Aurantii Fructus and their adulterants.

**Figure 6 molecules-27-04537-f006:**
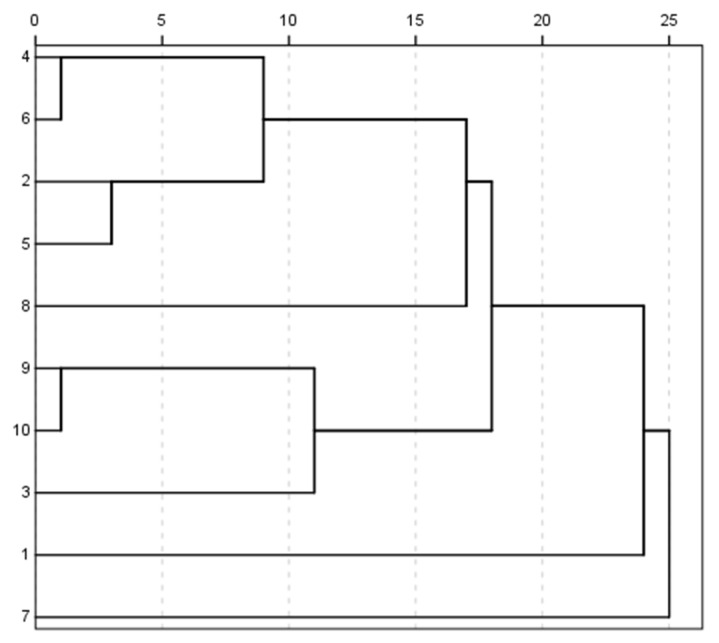
Cluster analysis of volatile compounds in Qu Aurantii Fructus and Aurantii Fructus. Note: X axis represents the classification distance, Y axis represents samples (1: *Citrus aurantium* L.; 2: *Citrus aurantium* ‘Huangpi’; 3: *Citrus aurantium* L.; 4~6: *Citrus aurantium* cv. Xiucheng; 7: *Citrus aurantium* ‘Daidai’; 8: *Citrus aurantium* ‘Chuluan’; 9~10: *Citrus changshan-huyou* Y.B.chang).

**Figure 7 molecules-27-04537-f007:**
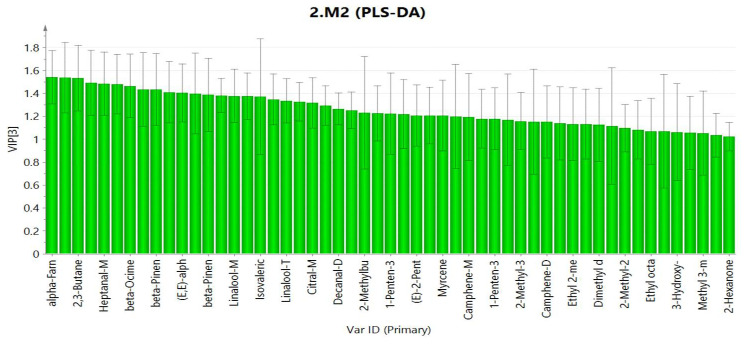
PLS-DA analysis of volatile compounds in Qu Aurantii Fructus, Aurantii Fructus and adulterants.

**Figure 8 molecules-27-04537-f008:**
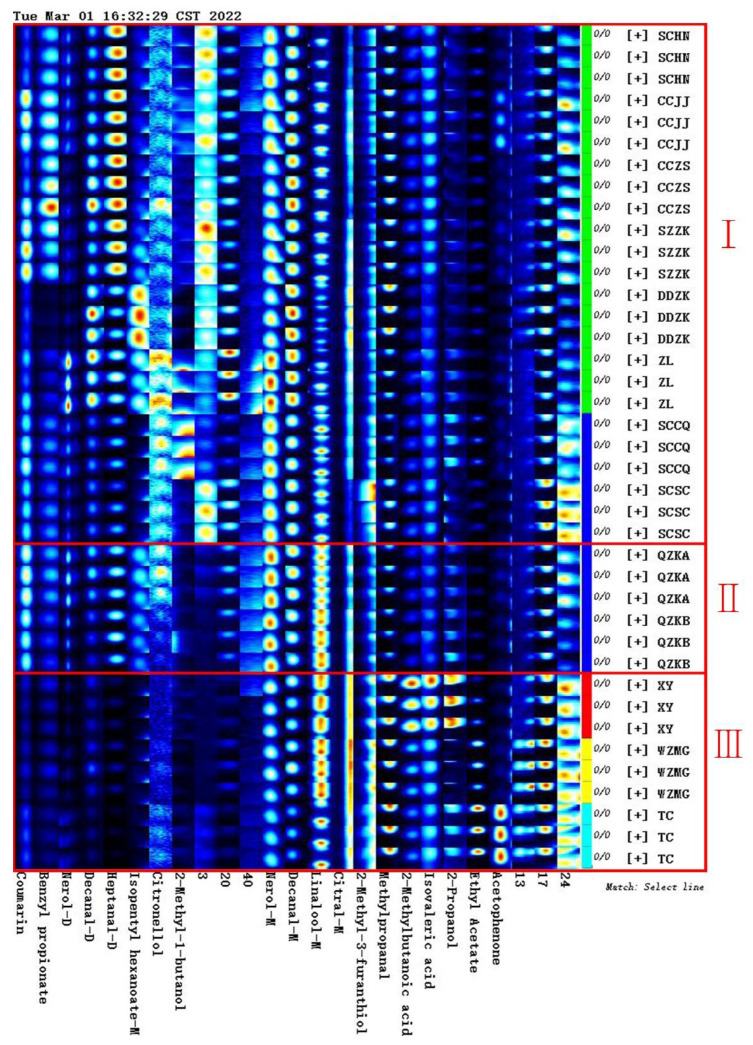
Fingerprint of characteristic components of Qu Aurantii Fructus. Note: each row represents sample data (from top to bottom: region I: Aurantii Fructus; region II: Qu Aurantii Fructus; region III: adulterants); each column represents a compound.

**Table 1 molecules-27-04537-t001:** The specific information and relative contents of volatile compounds in Qu Aurantii Fructus.

Compound	CAS	Formula	MW	RI	Dt (RIP Rel.)	Area Percentages (*n* = 8)	Range	Comment
limonene	138-86-3	C_10_H_16_	136.2	1025.2	1.68	6.93%	5.93–8.29%	monomer
limonene	138-86-3	C_10_H_16_	136.2	1026.2	2.17			dimer
α-farnesene	502-61-4	C_15_H_24_	204.4	1520.0	1.45	10.41%	6.88–14.46%	monomer
α-farnesene	502-61-4	C_15_H_24_	204.4	1551.4	1.43			dimer
γ-terpinene	99-85-4	C_10_H_16_	136.2	1066.9	1.21	6.24%	5.62–7.07%	monomer
γ-terpinene	99-85-4	C_10_H_16_	136.2	1065.6	1.70			dimer
linalool	78-70-6	C_10_H_18_O	154.3	1118.7	1.222	5.51%	4.20–8.20%	monomer
linalool	78-70-6	C_10_H_18_O	154.3	1117.3	1.76			dimer
linalool	78-70-6	C_10_H_18_O	154.3	1118.7	2.24			trimer
α-terpineol	98-55-5	C_10_H_18_O	154.3	1209.5	1.22	5.07%	4.30–5.84%	monomer
α-terpineol	98-55-5	C_10_H_18_O	154.3	1211.2	1.78			dimer
camphene	79-92-5	C_10_H_16_	136.2	959.8	1.64	4.50%	3.65–5.37%	monomer
camphene	79-92-5	C_10_H_16_	136.2	959.1	2.19			dimer
α-ocimene	13877-91-3	C_10_H_16_	136.2	1048.3	1.71	4.38%	2.62–5.91%	monomer
β-ocimene	13877-91-3	C_10_H_16_	136.2	1049.7	2.14			dimer
methyleugenol	93-15-2	C_11_H_14_O_2_	178.2	1436.2	1.47	4.19%	3.38–6.08%	
linalool oxide	60047-17-8	C_10_H_18_O_2_	170.3	1081.1	1.26	2.92%	1.56–3.70%	monomer
linalool oxide	60047-17-8	C_10_H_18_O_2_	170.3	1082.4	1.81			dimer
α-thujene	2867-05-2	C_10_H_16_	136.2	916.0	1.67	2.80%	2.23–3.20%	
nerol	106-25-2	C_10_H_18_O	154.3	1239.4	1.31	2.33%	1.62–3.85%	monomer
nerol	106-25-2	C_10_H_18_O	154.3	1238.4	1.75			dimer
β-pinene	127-91-3	C_10_H_16_	136.2	979.8	1.72	2.15%	1.93–2.33%	monomer
β-pinene	127-91-3	C_10_H_16_	136.2	982.1	2.17			dimer
linalyl acetate	115-95-7	C_12_H_20_O_2_	196.3	1337.0	1.22	2.09%	1.53–3.41%	monomer
linalyl acetate	115-95-7	C_12_H_20_O_2_	196.3	1337.4	1.69			dimer
linalyl acetate	115-95-7	C_12_H_20_O_2_	196.3	1338.2	1.89			trimer
tricyclene	508-32-7	C_10_H_16_	136.2	905.0	1.66	2.01%	1.01–2.55%	
α-terpinene	99-86-5	C_10_H_16_	136.2	1006.7	1.22	1.97%	1.22–2.55%	monomer
α-terpinene	99-86-5	C_10_H_16_	136.2	1009.3	1.72			dimer
terpinen-4-ol	562-74-3	C_10_H_18_O	154.3	1163.6	1.22	1.61%	1.12–2.49%	monomer
terpinen-4-ol	562-74-3	C_10_H_18_O	154.3	1164.2	1.72			dimer
(*Z*)-β-farnesene	28973-97-9	C_15_H_24_	204.4	1489.7	1.45	1.50%	0.76–2.18%	
coumarin	91-64-5	C_9_H_6_O_2_	146.1	1520.6	1.22	1.46%	0.79–3.05%	
2-methoxy-4-methylphenol	93-51-6	C_8_H_10_O_2_	138.2	1163.6	1.19	1.09%	0.53–1.94%	
geraniol	106-24-1	C_10_H_18_O	154.3	1267.9	1.22	1.08%	0.75–1.48%	
decanal	112-31-2	C_10_H_20_O	156.3	1261.4	1.55	1.06%	0.55–1.98%	monomer
decanal	112-31-2	C_10_H_20_O	156.3	1260.6	2.06			dimer
propan-2-one	67-64-1	C_3_H_6_O	58.1	485.7	1.12	0.96%	0.42–1.82%	
trans-p-menth-2-en-1-ol	29803-81-4	C_10_H_18_O	154.3	1137.0	1.70	0.95%	0.54–2.07%	
myrcene	123-35-3	C_10_H_16_	136.2	994.9	1.68	0.83%	0.16–1.32%	
γ-octalactone	104-50-7	C_8_H_14_O_2_	142.2	1298.5	1.31	0.80%	0.31–2.08%	monomer
γ-octalactone	104-50-7	C_8_H_14_O_2_	142.2	1299.3	1.80			dimer
acetic acid	64-19-7	C_2_H_4_O_2_	60.1	576.3	1.16	0.64%	0.39–0.98%	
α-pinene	80-56-8	C_10_H_16_	136.2	931.1	1.21	0.63%	0.18–0.96%	
2-methylprop-2-enal	78-85-3	C_4_H_6_O	70.1	581.9	1.21	0.57%	0.42–0.92%	
borneol	507-70-0	C_10_H_18_O	154.3	1184.3	1.90	0.57%	0.44–0.76%	
citral	5392-40-5	C_10_H_16_O	152.2	1309.3	1.05	0.55%	0.42–0.83%	monomer
citral	5392-40-5	C_10_H_16_O	152.2	1310.1	1.61			dimer
benzothiazole	95-16-9	C_7_H_5_NS	135.2	1229.2	1.16	0.49%	0.25–0.77%	
1-(furan-2-yl)ethanone	1192-62-7	C_6_H_6_O_2_	110.1	893.5	1.12	0.49%	0.21–0.69%	monomer
1-(furan-2-yl)ethanone	1192-62-7	C_6_H_6_O_2_	110.1	893.5	1.44			dimer
3-methylbut-2-enal	107-86-8	C_5_H_8_O	84.1	766.6	1.36	0.49%	0.32–0.65%	
3-methylbutyl hexanoate	2198-61-0	C_11_H_22_O_2_	186.3	1281.4	1.53	0.46%	0.20–1.10%	monomer
3-methylbutyl hexanoate	2198-61-0	C_11_H_22_O_2_	186.3	1280.5	2.15			dimer
ethyl octanoate	106-32-1	C_10_H_20_O_2_	172.3	1256.1	1.49	0.45%	0.16–0.61%	
methanol	67-56-1	CH_4_O	32	393.2	0.99	0.44%	0.05–0.86%	
furfural	98-01-1	C_5_H_4_O_2_	96.1	812.1	1.08	0.43%	0.32–0.63%	monomer
furfural	98-01-1	C_5_H_4_O_2_	96.1	814.2	1.33			dimer
methyl acetate	79-20-9	C_3_H_6_O_2_	74.1	537.0	1.19	0.42%	0.23–0.88%	
2-methylbutanal	96-17-3	C_5_H_10_O	86.1	657.6	1.40	0.31%	0.17–0.60%	
butanoic acid	107-92-6	C_4_H_8_O_2_	88.1	788.9	1.17	0.30%	0.13–0.57%	
propanal	123-38-6	C_3_H_6_O	58.1	526.6	1.15	0.28%	0.16–0.42%	
ethanol	64-17-5	C_2_H_6_O	46.1	441.8	1.05	0.27%	0.11–0.70%	monomer
ethanol	64-17-5	C_2_H_6_O	46.1	442.1	1.14			dimer
vanillin	121-33-5	C_8_H_8_O_3_	152.1	1408.6	1.27	0.27%	0.20–0.43%	
butan-2-one	78-93-3	C_4_H_8_O	72.1	582.7	1.25	0.27%	0.11–0.46%	
octanal	124-13-0	C_8_H_16_O	128.2	998.2	1.82	0.24%	0.04–0.93%	
isopentanol	123-51-3	C_5_H_12_O	88.1	715.4	1.25	0.24%	0.02–0.41%	monomer
isopentanol	123-51-3	C_5_H_12_O	88.1	716.7	1.50			dimer
benzaldehyde	100-52-7	C_7_H_6_O	106.1	945.4	1.15	0.24%	0.21–0.29%	monomer
benzaldehyde	100-52-7	C_7_H_6_O	106.1	946.7	1.47			dimer
acetophenone	98-86-2	C_8_H_8_O	120.2	1073.1	1.19	0.23%	0.15–0.31%	
butane-2,3-dione	431-03-8	C_4_H_6_O_2_	86.1	572.1	1.18	0.22%	0.17–0.26%	
propan-1-ol	67-63-0	C_3_H_8_O	60.1	493.4	1.18	0.21%	0.08–0.47%	
ethyl decanoate	110-38-3	C_12_H_24_O_2_	200.3	1411.3	1.61	0.21%	0.19–0.23%	
(*Z*)-dec-4-enal	21662-09-9	C_10_H_18_O	154.3	1191.1	1.34	0.20%	0.17–0.25%	
(methyldisulfanyl)methane	624-92-0	C_2_H_6_S_2_	94.2	726.4	0.99	0.18%	0.09–0.45%	
(*E*)-hex-2-en-1-ol	928-95-0	C_6_H_12_O	100.2	833.9	1.18	0.17%	0.02–0.57%	monomer
(*E*)-hex-2-en-1-ol	928-95-0	C_6_H_12_O	100.2	832.0	1.52			dimer
pentanoic acid	109-52-4	C_5_H_10_O_2_	102.1	892.7	1.22	0.14%	0.05–0.23%	
ethyl acetate	141-78-6	C_4_H_8_O_2_	88.1	609.7	1.34	0.14%	0.11–0.18%	
heptan-2-one	110-43-0	C_7_H_14_O	114.2	870.2	1.26	0.14%	0.05–0.26%	monomer
heptan-2-one	110-43-0	C_7_H_14_O	114.2	870.2	1.63			dimer
benzyl propionate	122-63-4	C_10_H_12_O_2_	164.2	1347.4	1.36	0.14%	0.08–0.29%	
methylpropanal	78-84-2	C_4_H_8_O	72.1	554.7	1.28	0.14%	0.06–0.20%	
4-methyl-3-penten-2-one	141-79-7	C_6_H_10_O	98.1	778.5	1.44	0.13%	0.03–0.19%	
hexanal	66-25-1	C_6_H_12_O	100.2	779.6	1.56	0.13%	0.03–0.41%	dimer
2-methylbutanoic acid	116-53-0	C_5_H_10_O_2_	102.1	879.9	1.20	0.12%	0.07–0.16%	
citronellol	106-22-9	C_10_H_20_O	156.3	1266.1	1.85	0.11%	0.09–0.14%	
heptanal	111-71-7	C_7_H_14_O	114.2	879.9	1.35	0.11%	0.03–0.37%	monomer
heptanal	111-71-7	C_7_H_14_O	114.2	882.1	1.70			dimer
ethyl propanoate	105-37-3	C_5_H_10_O_2_	102.1	693.4	1.45	0.11%	0.03–0.17%	
hexan-2-ol	626-93-7	C_6_H_14_O	102.2	766.6	1.29	0.10%	0.04–0.16%	
pent-1-en-3-one	1629-58-9	C_5_H_8_O	84.1	672.5	1.31	0.10%	0.02–0.28%	
(*E*)-hept-2-enal	18829-55-5	C_7_H_12_O	112.2	929.7	1.26	0.09%	0.06–0.13%	monomer
(*E*)-hept-2-enal	18829-55-5	C_7_H_12_O	112.2	942.3	1.67			dimer
3-methylbutanal	590-86-3	C_5_H_10_O	86.1	643.4	1.41	0.09%	<0.01–0.17%	
pentan-1-ol	71-41-0	C_5_H_12_O	88.1	748.1	1.26	0.08%	0.04–0.13%	
1-hydroxypropan-2-one	116-09-6	C_3_H_6_O_2_	74.1	640.2	1.22	0.08%	0.05–0.11%	
6-methyl-5-hepten-2-one	110-93-0	C_8_H_14_O	126.2	972.7	1.17	0.08%	0.02–0.18%	
ethyl benzoate	93-89-0	C_9_H_10_O_2_	150.2	1179.4	1.27	0.08%	0.05–0.13%	
1-penten-3-ol	616-25-1	C_5_H_10_O	86.1	678.0	1.34	0.07%	0.02–0.16%	
hexan-1-ol	111-27-3	C_6_H_14_O	102.2	855.0	1.33	0.07%	0.06–0.09%	monomer
hexan-1-ol	111-27-3	C_6_H_14_O	102.2	855.6	1.64			dimer
isovaleric acid	503-74-2	C_5_H_10_O_2_	102.1	879.9	1.23	0.07%	0.04–0.10%	
benzeneacetaldehyde	122-78-1	C_8_H_8_O	120.2	1028.1	1.26	0.06%	0.05–0.07%	
3-hydroxybutan-2-one	513-86-0	C_4_H_8_O_2_	88.1	702.5	1.33	0.06%	0.03–0.08%	
butanal	123-72-8	C_4_H_8_O	72.1	597.3	1.29	0.05%	0.02–0.06%	
pyrrole	109-97-7	C_4_H_5_N	67.1	743.2	0.97	0.05%	0.02–0.08%	
3-methylpentan-1-ol	589-35-5	C_6_H_14_O	102.2	831.3	1.60	0.04%	0.02–0.07%	
3-methylpentan-2-one	565-61-7	C_6_H_12_O	100.2	753.8	1.49	0.04%	0.03–0.05%	
hexan-2-one	591-78-6	C_6_H_12_O	100.2	768.7	1.20	0.04%	0.02–0.06%	monomer
hexan-2-one	591-78-6	C_6_H_12_O	100.2	768.2	1.51			dimer
(*E*)-pent-2-enal	1576-87-0	C_5_H_8_O	84.1	739.2	1.36	0.04%	0.02–0.10%	
3-methylsulfanylpropanal	3268-49-3	C_4_H_8_OS	104.2	887.8	1.09	0.04%	0.02–0.06%	
2-methylfuran-3-thiol	28588-74-1	C_5_H_6_OS	114.2	871.4	1.14	0.04%	0.01–0.09%	
2-oxopropyl acetate	592-20-1	C_5_H_8_O_3_	116.1	848.3	1.04	0.03%	<0.01–0.12%	
ethyl 2-methylbutanoate	7452-79-1	C_7_H_14_O_2_	130.2	853.2	1.23	0.03%	0.02–0.04%	
pentanal	110-62-3	C_5_H_10_O	86.1	688.3	1.42	0.03%	0.01–0.07%	
ethyl 2-methylpropanoate	97-62-1	C_6_H_12_O_2_	116.2	753.4	1.56	0.02%	<0.01–0.05%	
2-methylbutan-1-ol	137-32-6	C_5_H_12_O	88.1	736.7	1.48	0.02%	0.01–0.03%	
2-ethyl pyrazine	13925-00-3	C_6_H_8_N_2_	108.1	918.8	1.12	0.02%	0.01–0.04%	
(*E*)-2-methylpent-2-enal	623-36-9	C_6_H_10_O	98.1	818.3	1.49	0.02%	0.01–0.03%	
2,5-dimethylfuran	625-86-5	C_6_H_8_O	96.1	705.0	1.36	0.02%	0.01–0.03%	
propanol	71-23-8	C_3_H_8_O	60.1	540.0	1.24	0.02%	<0.01–0.04%	
furan-2-ylmethanol	98-00-0	C_5_H_6_O_2_	98.1	846.7	1.38	0.02%	0.01–0.03%	
(*Z*)-3-hexen-1-ol	928-96-1	C_6_H_12_O	100.2	844.0	1.52	0.02%	0.01–0.05%	
isopropyl acetate	108-21-4	C_5_H_10_O_2_	102.1	652.2	1.48	0.01%	0.01–0.03%	
isoamyl acetate	123-92-2	C_7_H_14_O_2_	130.2	854.9	1.75	0.01%	0.01%	
ethyl 2-methylbutanoate	7452-79-1	C_7_H_14_O_2_	130.2	826.9	1.65	0.01%	0.01%	
methyl 3-methylbutanoate	556-24-1	C_6_H_12_O_2_	116.2	756.1	1.53	0.01%	0.01	

Abbreviations: MW, Molecular weight; RI, retention index; Dt (RIP Rel.), drift time (reaction-ion-peak relative).

**Table 2 molecules-27-04537-t002:** Sample information table.

No.	Name	Species	Place of Origin
1	Qu Aurantii Fructus	*Citrus changshan-huyou* Y.B.Chang	Quzhou City, Zhejiang Province
2	Qu Aurantii Fructus	*Citrus changshan-huyou* Y.B.Chang	Quzhou City, Zhejiang Province
3	Qu Aurantii Fructus	*Citrus changshan-huyou* Y.B.Chang	Quzhou City, Zhejiang Province
4	Qu Aurantii Fructus	*Citrus changshan-huyou* Y.B.Chang	Quzhou City, Zhejiang Province
5	Qu Aurantii Fructus	*Citrus changshan-huyou* Y.B.Chang	Quzhou City, Zhejiang Province
6	Qu Aurantii Fructus	*Citrus changshan-huyou* Y.B.Chang	Quzhou City, Zhejiang Province
7	Qu Aurantii Fructus	*Citrus changshan-huyou* Y.B.Chang	Quzhou City, Zhejiang Province
8	Qu Aurantii Fructus	*Citrus changshan-huyou* Y.B.Chang	Quzhou City, Zhejiang Province
9	Aurantii Fructus	*Citrus aurantium* L.	Qijiang County, Sichuan Province
10	Aurantii Fructus	*Citrus aurantium* L.	Chongqing City
11	Aurantii Fructus	*Citrus aurantium* ‘Huangpi’	Yuanjiang City, Hunan Province,
12	Aurantii Fructus	*Citrus aurantium* cv. Xiucheng	Jiujiang City, Jiangxi Province
13	Aurantii Fructus	*Citrus aurantium* cv. Xiucheng	Zhangshu City, Jiangxi Province
14	Aurantii Fructus	*Citrus aurantium* cv. Xiucheng	Sanhu Town, Xingan County, Jiangxi Province
15	Aurantii Fructus	*Citrus aurantium* ‘Daidai’	Quzhou City, Zhejiang Province
16	Aurantii Fructus	*Citrus aurantium* ‘Chuluan’	Dongtou, Wenzhou City, Zhejiang Province
17	adulterants	*Citrus wilsonii* Tana-ka,	Hanzhong City, Shanxi Province
18	adulterants	*Citrus**reticulata* ‘Unshiu’	Wenzhou City, Zhejiang Province
19	adulterants	*Citrus**sinensis* (Linn.) Osbeck	Quzhou City, Zhejiang Province

## Data Availability

The data presented in this study are available on request from the corresponding author.

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
