# Peer review of "Development of the Volatile Fingerprint of Qu Aurantii Fructus by HS-GC-IMS"

_molecules, 2022, doi:10.3390/molecules27144537_

Round 1
Reviewer 1 Report
Authors propose an interesting analytical strategy in order to evaluate the volatile fractions of Qu Aurantii Fructus samples. The analytical approach combines the high separation efficiency of gas chromatography (GC) with the rapid and sensitive response of ion migration spectroscopy (IMS). The optimized procedure involved the analysis of the volatile components by headspace injection without enrichment and concentration, without organic solvent extraction, according to green analytical principles. In fully-automated manner, the headspace was collected and directly injected in GC system. The analytical strategy is very interesting but the manuscript must be revisited for the publication in Molecules. In my opinion, it cannot be qualified as an “original paper” in actual form.
First of all, English edition by a professional is highly recommended! In several sections, the manuscript lacks flow and, sincerely, it was painful to read. In the introduction section, particular attention should be also paid in the description of the headspace techniques for the analysis of volatile molecules in flavour samples and their advantage than to solvent-based extraction approaches. After, the Authors should emphasize the automation of the sampling procedure without too much human intervention. This aspect is fundamental in the development of the analytical procedure!
Other important aspect regards the monomer, dimer and trimer forms of several compounds. What do you mean? For instance, limonene is a monoterpene (C10H16) with a given CAS number, molecular weight and so on… Is it existing a dimer of limonene with identical CAS number, molecular weight, retention indices (RI), etc.? Please comment this fact.
Just in case they are useful for a future version or other work, let me make some additional suggestions, comments and recommendations:
Introduction section: “volatile oil” is uncorrected in opinion of this Reviewer. May be the Authors intended volatile fractions or essential oil...Please, modify the sentence.
Introduction section: “GC-MS/MS” the Authors should specify the acronym of gas chromatography coupled to triple quadrupole mass spectrometry.” Please, consider the suggestion!
Introduction section: “HS-GC-IMS”. The Authors should report in extensive manner headspace (HS) gas chromatography method coupled with ion-mobility-spectrometry. Also in this case, it is necessary to specify the acronym of HS-GC-IMS.
Introduction section: “Compared with GC-MS/MS, this method is more simple, rapid, exclusive and sensitive [20].” This sentence is completely inappropriate. You cannot compare a separation methodology (GC-MS/MS) with an extractive and separative technique (HS-GC-IMS).
Materials and Methods section: In opinion of this Reviewer, the sections 2.2 and 2.3 could be unify. In this section, several considerations must be done. First of all, the sampling of the volatile compounds has been optimized or it is referred to the literature? Please, define this aspect and eventually you report the reference. Also, the stationary phase of capillary column is not specified. Please, to report it and describe the provider. Why the Authors utilize a capillary column with a high internal diameter (0.53) for the analytical separation? Please, comment!
Figures: The figures are not evaluable due to their low-resolution.
Table 2: CAS numbers are not corrected. For instance, CAS number of the limonene is 138-86-3 and not C138863. Also, to describe the abbreviation in the caption section of the table. What do you mean RI?? This Reviewer image that the Authors refer to retention indices, but no experimental evidences are reported. Please, to report the methodology used in order to determine the RI of volatile components.
Reviewer 2 Report
In present manuscript, HS-GC-IMS was used to study the volatile compounds of Qu Aurantii Fructus, and a total of 174 peaks were detected. To compare the volatile compounds of Qu Aurantii Fructus with similar medical herb, Aurantii Fructus, and their common adulterants, principal component analysis (PCA) and cluster analysis (CA) were performed based on the signal intensity of all the detected peaks. Although the authors written that this study provides a novel method for rapid and effective identification of Qu Aurantii Fructus and a new dimension to recognize the relationship between Qu Aurantii Fructus and Aurantii Fructus, I am not convinced since the compounds identification in Table 2 is unclear and needs detail explanation. Before performing the statistical analysis there is need to explain in detail the obtained data and the compounds identification with respect to the further more detail remarks. The manuscript needs major revision with respect to the compounds identification and there are lot of doubts and questions related to the compounds listed in Table 2.
Particular remarks:
The names of the compounds from abstract and overall text (including the graphs) should be written with small letters and Greek symbols should be used instead of words „alpha, beta…“, the symbols for isomers E, Z should be written italic.
How is it possible that limonene monomer and limonene dimer have the same molecular formula C10H16 and farnesene monomer and dimer C15H24, as well as terpinene monomer and dimer C10H16, linalool monomer and dimer C10H18O, terpineol monomer and dimer C10H18O, camphene dimer C10H16, ocimene monomer and dimer C10H16, linalool oxide monomer and dimer C10H18O2 and more as is written in Table 2? In addition, how is it possible that the monomers and dimers have the same CAS number? There is need for detail explanation.
In Table 2 average contents should be replaced with Area percentages.
IUPAC names for aliphatic compounds should be presented in Table 2. E.g. hexan-2-ol, (E)-hept-2-enal…
Instead of Retention time there is need to report the retention indices relative to the series of n-alkanes. What is the purpose of Dt presentation in the table?
The names of the compounds in paragraph 3.2.1. should be corrected (small letters. IUPAC names…).
Round 2
Reviewer 2 Report
The authors improved the manuscript regarding to the coments, but English language should be improved more.
Author Response
Thank you for your patience in reviewing my manuscript again. We really appreciate your efforts in reviewing our manuscript. We have revised the manuscript accordingly. Our point-by-point responses are detailed below.
Point 1: The authors improved the manuscript regarding to the coments, but English language should be improved more.
Response 1: We are very sorry that the English expression of this manuscript is not standardized enough, and we have revised some of the content again and marked it in blue font. If this revision still does not meet your requirements, we hope we can be given the opportunity to revise them again. Thank you again for your hard work.
Reviewer 3 Report
According to the suggestions of the reviewers, the overall quality of the study was improved. In my opinion, the manuscript could be published after minor revision.
General comment:
1. Abstract, a total of 174 peaks were detected, 102 volatile organic compounds were identifiedàWhy only 102 components were identified? What about the other 72 peaks?
2. Page 4 n-ketones à “n” Should be written in italics.
3. The order of “materials and methods” and “results and discussion” needs to be adjusted according to the template guidelines of molecules.
Author Response
Thank you for your patience in reviewing my manuscript again. We really appreciate your efforts in reviewing our manuscript. We have revised the manuscript accordingly. Our point-by-point responses are detailed below.
Point 1: Abstract, a total of 174 peaks were detected, 102 volatile organic compounds were identifiedàWhy only 102 components were identified? What about the other 72 peaks?
Response 1: Thank you for your careful review, as many compounds produce dimers or trimers during ionization, the 102 compounds identified actually correspond to 131 chromatographic peaks, which are listed in Table 1, but this fact was not described in the Abstract. Therefore, we have made an addition in the Abstract. In addition, we consider that maybe we did not number these peaks in Table 1, which is not conducive to everyone's judgment, but considering that if we add another column in Table 1, too much content will make typesetting difficult, so we did not add it. If you find it necessary to add it, we will add it in the next revision. At present, 43 peaks have not been identified because the IMS database is not perfect. We think that the unidentified peaks do not affect the analysis of the results. Of course, it is necessary for us to make further improvement in the future research. Point 2: Page 4 n-ketones à “n” Should be written in italics.Response 2: We are grateful for the suggestion. We have changed the “n-ketones” in the manuscript to “n-ketones”.Point 3: The order of “materials and methods” and “results and discussion” needs to be adjusted according to the template guidelines of molecules.
Response 3: We are grateful for the suggestion. We apologize for not improving this part in the first revision, and now we have modified it as required. Due to the adjustment of the order, we have made appropriate changes to the first paragraph of the result part and marked it in blue font.
These are the main contents of my current revision, if this revision still does not meet your requirements, we hope we can be given the opportunity to revise them again. Thank you again for your hard work.